# Unprecedented Potential for Neural Drug Discovery Based on Self-Organizing hiPSC Platforms

**DOI:** 10.3390/molecules25051150

**Published:** 2020-03-04

**Authors:** Agustín Cota-Coronado, Jennifer C. Durnall, Néstor Fabián Díaz, Lachlan H. Thompson, N. Emmanuel Díaz-Martínez

**Affiliations:** 1Medical and Pharmaceutical Biotechnology, Center for Research and Assistance in Technology and Design of the State of Jalisco, CIATEJ A.C, 800 Normalistas, Colinas de La Normal, Guadalajara 44270, Jalisco, Mexico; agustin.cotacoronado@florey.edu.au; 2The Florey Institute of Neuroscience and Mental Health, 30 Royal Parade, Parkville, VIC 3052, Australia; jennifer.durnall@florey.edu.au; 3Departamento de Fisiología y Desarrollo Celular, Instituto Nacional de Perinatología, Mexico City 11000, Mexico; nfdiaz00@yahoo.com.mx

**Keywords:** hiPSCs, self-assembly organoids, high-throughput screening, 3D culture, drug-discovery platforms

## Abstract

Human induced pluripotent stem cells (hiPSCs) have transformed conventional drug discovery pathways in recent years. In particular, recent advances in hiPSC biology, including organoid technologies, have highlighted a new potential for neural drug discovery with clear advantages over the use of primary tissues. This is important considering the financial and social burden of neurological health care worldwide, directly impacting the life expectancy of many populations. Patient-derived iPSCs-neurons are invaluable tools for novel drug-screening and precision medicine approaches directly aimed at reducing the burden imposed by the increasing prevalence of neurological disorders in an aging population. 3-Dimensional self-assembled or so-called ‘organoid’ hiPSCs cultures offer key advantages over traditional 2D ones and may well be gamechangers in the drug-discovery quest for neurological disorders in the coming years.

## 1. Introduction

The generation of human induced pluripotent stem cells (hiPSCs) by Yamanaka and Takahashi revolutionized stem cell research—cell replacement therapy and drug discovery could now become personalized [1,2]. However a number of challenges have arisen, including those associated with cell replacement therapy such as the immaturity of the cells at the time of transplantation, the immune response they may elicit, the heterogeneity and reproducibility among worldwide laboratories handling patient-derived cell-lines [3]. Further, the reprogramming methods initially used to generate hiPSCs have been modified into non-integrative methods, making the new generations of hiPSCs ‘footprint free’ to curb concerns about the potential off-target effects of cellular reprograming [4]. hiPSCs can now be generated from a range of somatic cell types such as peripheral blood, keratinocytes and non-invasive sources as exfoliated renal cells and all can be potentially converted into functional neurons [5,6,7]. As hiPSCs technology has advanced, so too have differentiation protocols for the generation of neuronal cell types. Subtype-specific protocols with higher yields and purity have been described for many cell types, including dopaminergic [8] and GABAergic neurons [9], neural progenitors [10], astrocytes [11] and hippocampal granule cells [12]. Biological platforms that can mimic both temporal and cytoarchitectural aspects of human development, such as brain organoids, have emerged and will continue to expand in the field due to their enormous potential [13,14]. Therefore, hiPSCs and their in vitro differentiation are becoming a valid tool to understand human brain development (Figure 1), neurodevelopmental diseases, and to assess novel drug targets in human disease, perhaps reducing reliance on traditional animal modeling [15,16].

## 2. Human Induced Pluripotent Stem Cell (hiPSC)-Derived Region-Specific Neural Organoids

One key aspect of hiPSCs that has made them so attractive for research and drug development is that the underlying patient genetics are retained, including pathological mutations [15,17]. This allows for in vitro modelling of the pathological phenotype via their differentiation to the affected cell types [18,19]. Further, organoids have enabled the characterization of disease in more complex cellular milieus and this is of great interest for drug screening in that they may allow movement away from animal models, and perhaps an accelerated, more effective drug screening process for an individual’s disease [20,21].

Recent improvements in organoid culture have demonstrated several advantages over two-dimensional, monolayer differentiation (Table 1) [22,23]. Organoids can recapitulate discrete brain regions that arise during human brain development such as seen in cortical-plate [24], forebrain [25], midbrain [26] and hypothalamic organoids [27]. These self-assembly platforms can mimic some aspects of human brain development such as topological organization similar to human tissue and can even generate functionally mature brain cells that are synaptically connected [25,28]. As such these region-specific brain organoids are a promising in vitro approach to model brain development [29], understand neurodevelopmental diseases [30], and for personalized drug-screening when an individual’s hiPSCs are used [31,32].

Jo et al., generated midbrain organoids derived from hPSCs, that exhibited key features of the human midbrain such as expression of mature dopaminergic neurons, dopamine release, electrophysiological responses and interestingly, neuromelanin-producing neurons [26]. Brain-organoids could provide evidence of novel pathological mechanisms that in 2D cultures perhaps are missing. Patient hiPSCs harboring the LRRK2-G2019S mutation have been differentiated to human midbrain organoids to model familial Parkinson’s Disease (PD) [18]. Their findings included reduced numbers in midbrain-dopaminergic neurons and a significant increase in the Forkhead Box A2 (Foxa2) gene compared with healthy control hiPSCs, suggesting a direct pathophysiological consequence that can be modelled in hiPSCs-derived LRRK2-G2019S midbrain organoids [18]. Similarly, Kim et. al. found a pathological mechanism underlying the sporadic LRRK2-G2019S mutation. Here, TXNIP gene expression (which is a major regulator of cellular redox signaling protecting cells from oxidative stress) was upregulated in midbrain organoids carrying the mutation compared to isogenic controls. These results suggested a direct relationship between α-synuclein (a hallmark of PD pathogenesis) and *TXNIP* [39], offering a new drug target for the treatment of sporadic PD.

Cederquist et al. recently described the generation of a self-organizing forebrain organoid that mimicked the topographical organization of the developing human forebrain. A hiPSC line with induced Sonic-hedgehog (i-SHH) signaling instructed the positional identity of neural cells in a distance-dependent manner, recapitulating the in vivo topographical organization of the developing human forebrain. This was the first report of a self-organizing organoid with defined anterior-posterior, dorso-ventral, and medio-lateral positioning [25].

Region-specific brain organoids can now be generated on a relatively large scale and reproducibly [45,46]. Brain organoids were used effectively in the race to understand the pathology of Zika virus (ZIKV) and to screen for drugs to combat the outbreaks seen recently in Africa, and then elsewhere around the world. In addition to the high fatality rate of people infected with ZIKV, it was infecting pregnant women and causing microcephaly in their newborns. Qian et al., employed cortical organoids to study the mode of infection of ZIKV and the link to microcephaly, and to test drugs to stop infection. Cortical organoids recapitulate key features of human cortical development and this delivered the researchers a platform to study the microcephaly seen in the developing newborns of Zika-infected women. As in human brain development, cortical organoids feature a progenitor zone organization, neurogenesis, similar gene expression and notably, the formation of a distinct human-specific outer radial glia cell layer (not evolutionarily conserved in rodents). Qian et al. developed a method for high throughput, cost effective production of cortical organoids and exposed the organoids to the Zika virus. They observed a preferential infection for SOX2^+^ neural progenitors from the African and Asian ZIKV versus other neural cell types. Interestingly, they observed a reduction in proliferation and a decrease in the neuronal-cell layer volume, mimicking microcephaly [38]. Overall, the method enabled the use of cortical organoids as an efficient tool to understand the pathology of Zika virus and as a high-throughput drug-screening platform with significant reproducibility.

The delivery of drugs through the Blood-Brain Barrier (BBB) is a major challenge for effective delivery to the central nervous system (CNS) [47,48]. Various groups are developing more relevant human BBB models based on hiPSCs in conjunction with other adult human cells as endothelial cells and or pericytes. Ribecco-Lutkiewicz et al. developed a novel hiPSCs-derived BBB model comprised of induced brain endothelial cells (i-BEC), and hiPSCs-derived neurons and astrocytes that exhibited the correct gene and protein expression profile as well as functional, polarized BBB transport. In addition the i-BBB exhibited high Trans Endothelial Electrical Resistance (TEER) and showed receptor mediated transcytosis using species cross-reactive BBB-crossing antibodies [49]. Recent work by Bergmann et al. explained a detailed protocol to generate BBB-organoids to evaluate drug-permeability. The authors were able to generate a scaled in vitro platform in 3 days (BBB-organoids) suitable for drug HTS evaluation with high efficiency. The group evaluated the BBB-organoid through the small molecule phosphatidylinositol 3-kinase inhibitor BKM120 that can cross the BBB and another compound with limited penetration, dabrafenib. They observed high amounts of BKM120 in the BBB-organoid and the presence of dabrafenib was not detected, therefore demonstrating BBB selectivity [50]. BBB-organoids are highly useful platforms that can recapitulate the in vivo properties of the BBB permeability [51,52], with the potential to surpass the 5% of the current drugs that can actively access to the CNS, in order to increase the repertory of crossing actively compounds [53].

## 3. Novel Drug-Screening Approaches

The pre-clinical drug screening process is associated with large costs and there are two cruical areas that inflate these costs; the lack of human disease models and the efficient identification of relevant drug targets. Human iPSCs-derived cell types and region-specific organoids help to overcome the lack of human disease models. Their use has been propsed for pre-clinical research and should reduce the number of animals required for pre-clinical tests, and perhaps remove the need for animal testing in the future. These in vitro modeling systems are amenable to high throughput screening for drug compounds as methods for production become reproducible in large scale, such that in theory, organoids will enable more efficient identification of disease-modifying drugs. Analysis of these in vitro platforms can be done with a variety of current genomic technologies including next generation sequencing, single-cell RNA analysis, and epigenomics, generating a large amount of data that can be correlated to patient genetics, their disease onset and progression [54]. Further, off-target effects of drugs and compunds may be revealed in such human models that otherwise go undetected in animal models.

The use of artificial intellegence (AI) to identify compounds and drugs that are more likely to be disease-modifying and have outstanding activity, so-called “real hits”, is becoming more widely employed. AI utilises molecular virtual databases known as “chemical space” where the generation of drug compounds is estimated in 10^63^ [55], and this can be explored and used to guide lead optimization programmes for ensuring biological activity [56]. AI and advanced computational techniques are key to the restriction of compounds that are truly relevant for human therapeutic use. One example was describe by Klingler et al., where they introduced a concept for accelerated discovery of structure-activity relationship (SAR) for enrichment within the large chemical space. The space navigation is accomplished within minutes on affordable standard computer hardware using a tree-based molecule descriptor and dynamic programming [57]. The approach is significantly timesaving, taking advantage of the “chemical space”, accelerating hit generation and huge optimization in drug design.

In another recent work, Naveja and Medina-Franco introduced an approach for better analyzing of the chemical space, organizing compounds in analog series of groups called “constellations”, this allowed a better identification in large datasets of unnoticed compounds as in HTS assays. They demonstrated the utility of their method using two datasets of inhibitors against DNA methyltransferases (DNMTs) and Serine/Threonine kinase 1 (AKT1). The method identified compounds with activity not previously reported through the categorizing and visualization of the constellation’s clusters [56]. This approach allowed a faster visualization of novel biological activity and physicochemical properties in the compounds, relevant for academia and industry. Overall, the new drug-discovery approaches ensure more target accuracy, nonetheless, the next step requires validation in human derived cells. Therefore, combination with 2D and 3D hiPSCs cultures could lead to a drug-screening revolution in terms of hit-relevancy and could shorten clinical acceptance process of novel compounds (Figure 2). This is further reinforced by the recent advances in AI and machine learning, due the enormous potential throughout the pharmaceutical hunting, making the process, cheaper and more effective [58]. Few companies have already started validation the effectiveness of AI in developing algorithms looking for drug-structure patterns in curate databases and research papers. Interestingly, the company BenevolentBio (Brooklyn, New York, US) is currently using AI for finding new ways to treat Amyotrophic Lateral Sclerosis (ALS), also known as Motor Neuron Disease (MND), it defined 100 compounds and from there the researchers selected 5 and they tested in patient-derived neurons, finally they found prominent activity in one of them, slowing neurological symptoms of the disease in mice [58].

The prior example explains the feasibility of testing new compounds predicted by AI in hiPSCs-derived neurons, while other groups demonstrated the utility of hiPSCs derived neurons as platforms for testing neurite growth related drugs, which is highly relevant for neural repair [34]. In this context, high throughput screening of drugs on human iPSC differentiated neurons proved useful for assessing toxicity of compounds, determining off-target effects of compounds already in use and to identify new mechanisms and specific molecular pathways to target. Sherman and Bang, assessed neurite growth-related drugs and from over 4000 small active compounds they identified 108 hits, confirming previous compounds and pathways associated to neuritogenesis, but they also identified novel compounds as well. Among the new findings: 2-methoxyphenylacryloyllupinine, two Chinese herbal medicines—diterpines and andrographolide—and the smooth muscle relaxant alverine citrate were found to increase neuritogenesis in induced cortical-like cells. Triptolide, identified as a hit, was found to inhibit neurite growth, which could be useful to inhibit pathogenic outgrowth of human neurons [34].

Sridharan et al., developed another robust platform for HTS screening in hiPSCs-derived induced neurons (iNs) through the viral transduction of the pro-neural gene NGN2. They simplified the overall process to increase scalability of iNs, including the generation of large batches of cryopreserved neurons. Post-thaw these iNs were tested in a phenotypic toxicity assay with the LOPAC library (1280 bioactive small molecules) identifying 14 compounds that targeted neurite outgrowth [35]. This work is a prominent example of generating robust and efficient protocols for HTS assays of hiPSCs-derived neurons.

## 4. Challenges and Perspectives

One of the major hurdles associated with hiPSCs neural differentiation is the inherent heterogeneity of the culture. To overcome this Zhang et al. developed a method to generate nearly 100% purity of excitatory neurons which were relatively mature within two weeks of culture via the forced overexpression of the pro-neural gene Neurogenin-2 (NGN2) from both hESCs and hiPSCs [59]. Kondo et al. used the same approach to obtain highly yields (nearly 100%) of cortical neurons with hiPSCs derived from familial and sporadic Alzheimer’s patients, providing a novel platform for new drug development [35]. As mentioned above, Sridharan et al. used the same protocol for rapid glutamatergic induced-neurons (iNs) production with high purity [35]. These and other approaches, including the evaluation of the transcriptional identity markers through fluorescent reporters during differentiation protocols and Fluorescent Activate Cell-Sorting (FACS) may significantly improve homogeneity of hiPSCs cultures [60].

One of the most promising approaches in recent years is the development of hiPSCs-derived neuronal cultures that can ‘self-assemble’ within microfluidic devices and therefore promote neurite outgrowth and interaction with other neural cell-types and enhance synaptic connections [61]. These so-called “organs-in-a-chip” are set to revolutionize drug-discovery [62,63,64]. Recent work published by Park et al. developed a microfluidic BBB-chip model from hiPSCs with which the group could mimic the conditions of hypoxia during brain development [65]. By generating hiPSCs-derived brain microvascular endothelial cells (i-BMVECs) and adding primary human pericytes and astrocytes, Park et al., produced a functional in vitro BBB that maintains relevant human physiological features for a week, presented permeability restriction that lasts up to 2 weeks, had high levels of expression of tight junction proteins, and appropriate function of efflux proteins. The group demonstrated that the BBB-chip was capable of transporter-mediated drug efflux including appropriate substrate specificity and they tested CNS-targeting peptides, nanoparticles, and antibodies crossing the BBB, demonstrating the BBB-chip could test clinically relevant compounds [65].

Therefore, regional organoids and organs-in-a-chip, recapitulate more closely the developmental course of differentiation of the diverse neurological populations and mimic the architectural interaction within them, in comparison with 2D cultures [13,16]. Nonetheless, they are emerging technologies with several limitations at this moment. Despite the enormous potential in disease modeling and drug screening, a more in-depth study revealed key differences between the organoids and normal human cortex development. Bhaduri et al. [66] tested the fidelity of cortex organoids in the generation of spatio-temporal diverse cell types, accompanied by their transcriptional signatures. They performed high-throughput single-cell RNA sequencing at different time-points of the developing human cortex (6-22 gestational weeks) obtained from seven different regions; prefrontal (PFC), motor, parietal, somatosensory and primary visual (V1) cortices as well as hippocampus and from 37 cortical organoids, comparing them with published single-cell cortical organoids datasets. Their findings revealed a smaller number of cell subtypes in the cortical organoids, and interestingly, the co-expression of pan-radial glia and pan-neuronal markers, instead of mature well-defined signatures. On the other hand, they found that the organoid environment activated genes as *PGK1* related to glycolysis and augmentation in ER stress through the genes *ARCN1* and *GORASP2,* regardless the stage of differentiation. In addition, they transplanted 8-week old cortical organoids in post-natal day four mice and after 5 weeks they observed reduced cellular stress markers. Therefore, probing that the organoids conditions trigger stress signaling pathways, impairing the correct specification of neuronal and glial subtypes [66]. This extensive work demonstrated that current in vitro cortical organoids are limited in their ability to completely model human brain development, however important achievements have been made in the obtention of cellular diversity of primary cell types, which is really useful in terms of drug HTS assays of patient-derived cell lines.

Together, recent advances in the generation of region-specific brain organoids that recapitulate some features of human brain development closer than traditional 2D culture and the incorporation of microfluidic devices, as physiologically-relevant BBB models, has made HTS for novel drug discovery a viable pre-clinical platform that allows for the advancement of personalized medicine and may reduce the need for animal model testing, and thus decrease the great economic cost associated with traditional drug testing methods. Simultaneously, AI is gaining more relevancy in the drug-discovery quest and paving the way to make the overall process more efficient, less costly and more accurate.

## Figures and Tables

**Figure 1 molecules-25-01150-f001:**
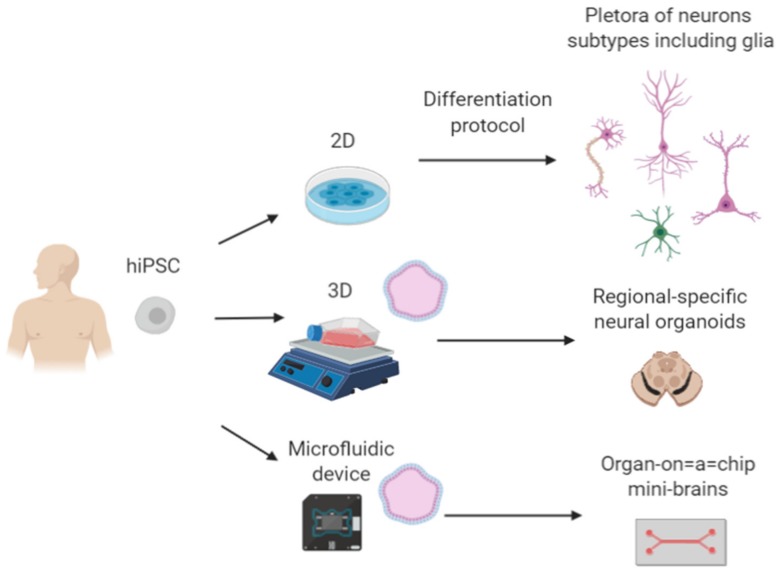
Self-assembly platforms obtained from hiPSCs-neurons. The refinement of induction protocols for neuronal subtypes has been critical for better outcomes in neural transplantation and drug-discovery. Therefore, the evolution towards 3D systems may enhance our understanding of neural development and generation of better disease models with more human relevancy. Finally, improvements in “organs-on-a-chip” technology could revolutionize drug-discovery based on precise medicine.

**Figure 2 molecules-25-01150-f002:**
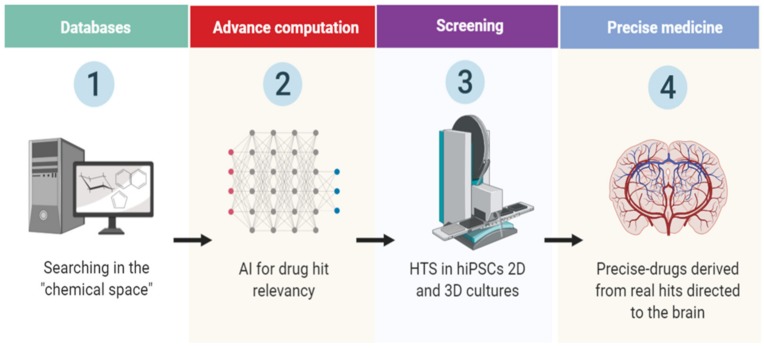
Flowchart of novel drug discovery in neurological disorders. The chemical space can be exploited by artificial intelligence (AI) advanced computational programs leading to better outcomes in drug hit discovery, performed in hiPSCs 2D–3D neuronal platforms. Therefore, the screening of novel targets is a highly dynamic process, based in the testing of thousands of compounds throughout different developmental stages in vitro.

**Table 1 molecules-25-01150-t001:** Assessment of the advantages of 2D, brain-organoids and Organs-on-a-chip for neural drug discovery. Differences between cell culture technologies were marked with symbols, which the higher score is four crosses and the less two.

Features	2D	Brain-Organoids	Organ-in-a-Chip	Ref
HTS screening assays	++++	+++	++++	[33,34,35]
Disease modeling	++	+++	+++	[18,36,37,38,39]
Mimic human neural development	++	++++	+++	[16,40,41,42]
Synaptic connection	++	+++	++++	[43,44]

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
