# Peer review of "Unprecedented Potential for Neural Drug Discovery Based on Self-Organizing hiPSC Platforms"

_molecules, 2020, doi:10.3390/molecules25051150_

Round 1

Reviewer 1 Report

The authors reviewed those application potentials of Human iPSCs which is developed into organoids and organ-on-a chips over recent years for the discovery tools of new drugs. This article presents the latest technology development in detail and is a well-written review paper.

Author Response

This reviewer didn’t suggest any changes..

Reviewer 2 Report

In this review, authors present a summary of the potential use of human iPSC self-organizing platforms for neurological diseases drugs discovery.

Although manuscript refers to a very relevant topic, in general, is not well written being very confused and disorganized. It is like a bombardment of ideas. This reviewer thinks that this review must be rewritten (especially points 3 and 4) presenting information in a more readable way. The inclusion of an additional table containing the most relevant information and references related to point 2 could be very useful and clarifying.

Minor points:

Line 33, non-integrative methods instead non-integration methods. Lines 199-204. Nowadays it is very difficult to avoid the need of using animal model testing. That is overstating reality, please soften this sentence.

Author Response

Answer:

We re-wrote the points 3 and 4, we agree that it was disorganized and we added more information for making the points more clear and readable.

  1. Novel drug-screening approaches

The pre-clinical drug screening process is associated with large costs and there are two cruical areas that inflate these costs;  the lack of human disease models and the efficient identification of relevant drug targets. Human iPSCs-derived cell types and region-specific organoids help to overcome the lack of human disease models. Their use has been propsed for pre-clinical research and should reduce the number of animals required for pre-clinical tests, and perhaps remove the need for animal testing in the future. These in vitro modeling systems are amenable to high throughput screening for drug compounds as methods for production become reproducible in large scale, such that in theory, organoids will enable more efficient identification of disease-modifying drugs. Analysis of these in vitro platforms can be done with a variety of current genomic technologies including next generation sequencing, single-cell RNA analysis, and epigenomics, generating a large amount of data that can be correlated to patient genetics, their disease onset and progression[54]. Further, off-target effects of drugs and compunds may be revealed in such human models that otherwise go undetected in animal models.

The use of artificial intellegance (AI) to identify compounds and drugs that are more likely to be disease-modifying and have outstanding activity, so-called “real hits”, is becoming more widely employed. AI utilises molecular virtual databases known as “chemical space” where the generation of drug compounds is estimated in 1063 [55], and this can be explored and used to guide lead optimization programmes for ensuring biological activity [56]. AI and advanced computational techniques are key to the restriction of compounds that are truly relevant for human therapeutic use. One example was describe by Klingler et al., (2019), where they introduced a concept for accelerated discovery of structure-activity relationship (SAR) for enrichment within the large chemical space. The space navigation is accomplished within minutes on affordable standard computer hardware using a tree-based molecule descriptor and dynamic programming [57]. The approach is significantly timesaving, taking advantage of the “chemical space”, accelerating hit generation and huge optimization in drug design.

In another recent work, Naveja and Medina-Franco., (2019), introduced an approach for better analyzing of the chemical space, organizing compounds in analog series of groups called “constellations”, this allowed a better identification in large datasets of unnoticed compounds as in HTS assays. They demonstrated the utility of their method using two datasets of inhibitors against DNA methyltransferases (DNMTs) and Serine/Threonine kinase 1 (AKT1). The method identified compounds with activity not previously reported through the categorizing and visualization of the constellation’s clusters [56]. This approach allowed a faster visualization of novel biological activity and physicochemical properties in the compounds, relevant for academia and industry. Overall, the new drug-discovery approaches ensure more target accuracy, nonetheless, the next step requires validation in human derived cells. Therefore, combination with 2D and 3D hiPSCs cultures could lead to a drug-screening revolution in terms of hit-relevancy and could shorten clinical acceptance process of novel compounds (Figure 2). This is further reinforced by the recent advances in AI and machine learning, due the enormous potential throughout the pharmaceutical hunting, making the process, cheaper and more effective [58]. Few companies have already started validation the effectiveness of AI in developing algorithms looking for drug-structure patterns in curate databases and research papers. Interestingly, the company BenevolentBio is currently using AI for finding new ways to treat Amyotrophic Lateral Sclerosis (ALS), also known as Motor Neuron Disease (MND), it defined 100 compounds and from there the researchers selected 5 and they tested in patient-derived neurons, finally they found prominent activity in one of them, slowing neurological symptoms of the disease in mice [58].  

The prior example explain the feasibility of testing new compounds predicted by AI in hiPSCs-derived neurons, while other groups demonstrated the utility of hiPSCs derived neurons as platforms for testing neurite growth related drugs, which is higly relevant for neural repair [34]. In this context, high throughput screening of drugs on human iPSC differentiated neurons proved useful for assessing toxicity of compounds, determining off-target effects of compounds already in use and to identify new mechanisms and specific molecular pathways to target. Sherman and Bang., (2018), assessed neurite growth-related drugs and from over 4,000 small active compounds they identified 108 hits, confirming previous compounds and pathways associated to neuritogenesis, but they also identified novel compounds as well. Among the new findings: 2-methoxy-phenylacryloyl-lupinine, two Chinese herbal medicines; Diterpines, Andrographolide, and the smooth muscle relaxant Alverine Citrate were found to increase neuritogenesis in induced cortical-like cells. Triptolide, identified as a hit, was found to inhibit neurite growth, which could be useful to inhibit pathogenic outgrowth of human neurons [34].

Sridharan et al., (2019), developed another robust platform for HTS screening in hiPSCs-derived induced neurons (iNs) through the viral transduction of the pro-neural gene NGN2. They simplified the overall process to increase scalability of iNs, including the generation of large batches of cryopreserved neurons. Post-thaw these iNs were tested in a phenotypic toxicity assay with the LOPAC library (1,280 bioactive small molecules) identifying 14 compounds that targeted neurite outgrowth [35]. This work is a prominent example of generating robust and efficient protocols for HTS assays of hiPSCs-derived neurons.

  1. Challenges and perspectives

One of the major hurdles associated with hiPSCs neural differentiation is the inherent heterogeneity of the culture. To overcome this Zhang et al., (2013) developed a method to generate nearly 100% purity of excitatory neurons which were relatively mature within two weeks of culture via the forced overexpression of the pro-neural gene Neurogenin-2 (NGN2) from both hESCs and hiPSCs [59]. Kondo et al., (2017) used the same approach to obtain highly yields (nearly 100%) of cortical neurons with hiPSCs derived from familial and sporadic Alzheimer’s patients, providing a novel platform for new drug development [35]. As mentioned above, Sridharan et al., (2019) used the same protocol for rapid glutamatergic induced-neurons (iNs) production with high purity [35]. These and other approaches, including the evaluation of the transcriptional identity markers through fluorescent reporters during differentiation protocols and Fluorescent Activate Cell-Sorting (FACS) may significantly improve homogeneity of hiPSCs cultures [60]. 

One of the most promising approaches in recent years is the development of hiPSCs-derived neuronal cultures that can ‘self-assemble’ within microfluidic devices and therefore promote neurite outgrowth and interaction with other neural cell-types and enhance synaptic connections [61]. These so-called “organs-in-a-chip” are set to revolutionize drug-discovery [62; 63; 64]. Recent work published by Park., et al (2019), developed a microfluidic BBB-chip model from hiPSCs with which the group could mimic the conditions of hypoxia during brain development [65]. By generating hiPSCs-derived brain microvascular endothelial cells (i-BMVECs) and adding primary human pericytes and astrocytes, Park et al, produced a functional in vitro BBB that maintains relevant human physiological features for a week, presented permeability restriction that lasts up to 2 weeks, had high levels of expression of tight junction proteins, and appropriate function of efflux proteins. The group demonstrated that the BBB-chip was capable of transporter-mediated drug efflux including appropriate substrate specificity and they tested CNS-targeting peptides, nanoparticles, and antibodies crossing the BBB, demonstrating the BBB-chip could test clinically relevant compounds [65].

Therefore, regional organoids and organs-in-a-chip, recapitulate more closely the developmental course of differentiation of the diverse neurological populations and mimic the architectural interaction within them, in comparison with 2D cultures [13; 16]. Nonetheless, they are emerging technologies with several limitations at this moment. Despite the enormous potential in disease modeling and drug screening, a more in-depth study revealed key differences between the organoids and normal human cortex development. Bhaduri et al., (2020) [66] tested the fidelity of cortex organoids in the generation of spatio-temporal diverse cell types, accompanied by their transcriptional signatures. They performed high-throughput single-cell RNA sequencing at different time-points of the developing human cortex (6-22 gestational weeks) obtained from 7 different regions; prefrontal (PFC), motor, parietal, somatosensory and primary visual (V1) cortices as well as hippocampus and from 37 cortical organoids, comparing them with published single-cell cortical organoids datasets. They’re findings revealed smaller number of cell subtypes in the cortical organoids, and interestingly, the co-expression of pan-radial glia and pan-neuronal markers, instead of mature well-defined signatures. On the other hand, they found that the organoid environment activated genes as PGK1 related to glycolysis and augmentation in ER stress through the genes ARCN1 and GORASP2, regardless the stage of differentiation. In addition, they transplanted 8-week old cortical organoids in post-natal day four mice and after 5 weeks they observed reduced cellular stress markers. Therefore, probing that organoids conditions trigger stress signaling pathways, impairing the correct specification of neuronal and glial subtypes [66]. This extensive work demonstrated that current in vitro cortical organoids are limited in their ability to completely model human brain development, however important achievements have been made in the obtention of cellular diversity of primary cell types, which is really useful in terms of drug HTS assays of patient-derived cell lines.    Together, recent advances in the generation of region-specific brain organoids that recapitulate some features of human brain development closer than traditional 2D culture and the incorporation of microfluidic devices, as physiologically-relevant BBB models, has made HTS for novel drug discovery a viable pre-clinical platform that allows for the advancement of personalized medicine and may reduce the need for animal model testing, and thus decrease the great economic cost associated with traditional drug testing methods. Alongside, AI is gaining more relevancies in the drug-discovery quest and paving the way to make the overall process more efficient, less costly and accurate.

We considered that is not that necessary to add another table with the new changes done. But if you consider that is mandatory, we can do it.

Line 33, non-integrative methods instead non-integration methods. Lines 199-204. Nowadays it is very difficult to avoid the need of using animal model testing. That is overstating reality, please soften this sentence

Answer: previous Line 33 non-integrative methods was changed by Line 34.- non-integration methods

We remove the phrase of Lines 199-204.

Reviewer 3 Report

Comment 1:

I think in general the paper is well organized and interesting but I would suggest discussing deeper the impact of human iPSCs technology in the field of drug discovery/testing with particular regard to organoids as a new and attractive platform for disease modeling and drug screening. Since this is a review I think it should be expanded in its discussion to be more easily approachable for people that are not familiar with iPSCs and their applications. 

Comment 2:

I would suggest a moderate English editing.

Author Response

Answer: We did agree in discuss more the relevancy of iPSCs in drug discovery and organoids, therefore we added more discussion and information regarding the drug screening approaches in hiPSCs and organoids platforms and a perspective how this 3D novel culturing techniques are increasing our knowledge of brain development and disease onset. In addition we mention how this technology is really promising, but not mature enough to completely mimic de development of complex regions of the brain as the human cortex. Overall, we cite examples in the points 3 and 4 of the paper, regarding the reviewer suggestions.

Round 2

Reviewer 2 Report

Authors have answered the issues raised by this reviewer in a satisfactory way. However, there are typos that must be corrected. 

Minor points:

Line 143, “proposed” instead of “propsed”

Line 151, “compounds” instead of “compunds”

Line 153 , “artificial intelligence” instead of “artificial intellegance”

Line 213, “The prior example explains” instead of “the prior example explain”

Line 284, “Their findings” instead of “They’re findings”